# Amino Acid Metabolism in Leukocytes Showing In Vitro IgG Memory from SARS-CoV2-Infected Patients

**DOI:** 10.3390/diseases12030043

**Published:** 2024-02-23

**Authors:** Giuseppina Fanelli, Veronica Lelli, Sara Rinalducci, Anna Maria Timperio

**Affiliations:** Department of Ecological and Biological Sciences, University of Tuscia, 01100 Viterbo, Italy; giuseppina.fanelli@unitus.it (G.F.); v.lelli@unitus.it (V.L.); sara.r@unitus.it (S.R.)

**Keywords:** SARS-CoV2, COVID-19, cell ELISA, metabolomics, mass spectrometry

## Abstract

The immune response to infectious diseases is directly influenced by metabolic activities. COVID-19 is a disease that affects the entire body and can significantly impact cellular metabolism. Recent studies have focused their analysis on the potential connections between post-infection stages of SARS-CoV2 and different metabolic pathways. The spike S1 antigen was found to have in vitro IgG antibody memory for PBMCs when obtaining PBMC cultures 60–90 days post infection, and a significant increase in S-adenosyl homocysteine, sarcosine, and arginine was detected by mass spectrometric analysis. The involvement of these metabolites in physiological recovery from viral infections and immune activity is well documented, and they may provide a new and simple method to better comprehend the impact of SARS-CoV2 on leukocytes. Moreover, there was a significant change in the metabolism of the tryptophan and urea cycle pathways in leukocytes with IgG memory. With these data, together with results from the literature, it seems that leukocyte metabolism is reprogrammed after viral pathogenesis by activating certain amino acid pathways, which may be related to protective immunity against SARS-CoV2.

## 1. Metabolic Rewiring in SARS-CoV2-Infected Patients

Severe Acute Respiratory Syndrome Coronavirus 2 (SARS-CoV2) is one of the seven human-infecting coronaviruses identified to date. It is a single-stranded positive-sense (+) ssRNA virus belonging to the β-coronavirus strain B1-3. In March 2020, the World Health Organization (WHO) declared the 2019 coronavirus disease caused by SARS-CoV2 (COVID-19) to be a pandemic, and according to the WHO over 762 million confirmed cases and over 6.8 million deaths have been reported worldwide (as of 9 April 2023) [1,2].

The clinical presentation of COVID-19 includes a wide spectrum of non-specific symptoms like fever, dry cough, dyspnea, headaches, sputum production, hemoptysis, myalgia, fatigue, nausea, vomiting, diarrhea, and abdominal pain [3,4]. Loss of smell and taste have not been commonly described in China; however, they have more recently been described as an early clinical marker for COVID-19 [5,6,7,8]. COVID-19 patients can be classified as asymptomatic or symptomatic and symptoms can range from mild to severe and critical [9].

Among all SARS-CoV2 variants identified, Alpha (B.1.1.7), Delta (B.1.617.2), and Omicron (B.1.1.529/BA.1) have been considered worldwide as variants of concern (VOCs), in addition to the historical (20A.EU2) variant that originated in Wuhan, China. [10]

Despite the similar symptoms, the Delta variant was found to be more transmissible than Alpha [11]. In contrast, the Omicron variant has shown high infectivity but causes less severe symptoms than previous variants [12]. People who have risk factors, such as advanced age, smoking, and concomitant diseases (diabetes, hypertension, cardiovascular disease, obesity, chronic lung disease, kidney disease) are more susceptible to severe acute respiratory syndrome (SARS) [13].

Once infected, humans transmit the virus through droplets, sneezing, and aerosols [14]. Some results in the literature suggest that patients with asymptomatic or only mild symptoms may excrete large amounts of the virus in the initial phase of infection, which promotes the rapid spread of the virus, although they are less contagious than patients with symptoms [15,16,17].

SARS-CoV2 elicits both innate and specific immune responses. In this regard, Ni et al. evaluated the SARS-CoV2-specific humoral and cellular immunity of patients who had recovered and discovered that both B and T cells play a role in protecting against viral infection through the immune response [18]. In this regard, accurate tests that can identify the SARS-CoV2 virus and monitor antiviral antibodies are essential for identifying those who have had an immune response to the virus, aiding in virus management and surveillance and combating the COVID-19 pandemic [19]. Furthermore, understanding the molecular mechanism involved in physiological recovery from viral infections and immune activities in leukocytes [20] became crucial for knowing the antibody-mediated immune response to SARS-CoV2.

The S1 protein (S1 protein) on the surface of SARS-CoV2 is the target of immune responses triggered by this viral infection [21,22]. Immune responses against infectious diseases are directly influenced by metabolic activities. Several studies have concentrated their analysis on the potential relationships between the post-infection stages of SARS-CoV2 and different metabolic pathways [23,24] in order to gain a better understanding of the biochemistry behind the leukocyte response [25].

Although not well understood, host defense mechanisms against SARS-CoV2 infection are crucial to the progression and potential outcome of the disease. Metabolic profiling can add to the lack of knowledge about the molecular mechanisms underlying the clinical manifestations and pathogenesis of COVID-19 to find robust diagnostic and prognostic biomarkers and to investigate the natural historical course of the infection [26,27]. In addition, the early identification of metabolomics-based biomarker signatures has been shown to be an effective approach for predicting disease progression [28]. Indeed, metabolomics is considered a hallmark approach in understanding disease pathophysiology and may play an essential role in the analysis of factors associated with gene function, transcription, mRNA degradation, post-translation modification, metabolite concentrations, and other cell activity processes associated with disease [29], and it has provided a powerful platform for revealing the molecular mechanisms underlying the pathogenesis of infections with early SARS-CoV2 strains [30].

Among metabolic reprogramming, amino acid metabolism plays a significant part in the regulation of immune responses and the assembly of progeny viruses, and alternations in amino acid metabolism over the course of SARS-CoV2 infection has become one of the focuses in COVID-19 research [31,32,33,34].

The amino acids can exert regulatory effects at various levels of cell activity, acting as mediators or signal molecules and, therefore, modulating numerous functions, allowing for the adequate expression of these regulatory activities in vivo during inflammatory states [35].

In fact, Maltais-Payette and colleagues investigated the association between COVID-19-symptom severity and circulating amino-acid concentrations and found that symptom severity is correlated to the concentration of 16 out of 20 amino acids. In particular, severe COVID-19 is positively associated with the concentrations of leucine, isoleucine, valine, glutamate, and phenylalanine, whereas glutamine, tryptophan, histidine, alanine, proline, and cysteine have an opposite effect [32]. A distinct investigation examined the metabolome of COVID-19 patients to recognize variations that are associated with the severity of the illness and to identify metabolites or metabolomic profiles that can serve as predictive markers. The results showed that, as the disease severity advanced, serum phenylalanine concentrations increased significantly, indicating that amino acid metabolism was impacted in COVID-19 patients. In contrast, levels of alanine, citrulline, and proline decreased as the disease worsened [36]. Additionally, metabolomics analysis revealed a modified amino acid metabolism correlating with altered oxygen homeostasis in COVID-19 patients [37].

In the report of Fanelli et al. [20], we provide a description of metabolites that describe the later phases of immune defense against SARS-CoV2 infection, which involve the absence of circulating antibodies but the presence of antibody memory. We examined the metabolic profiles of 41 cell cultures of peripheral blood mononuclear cells (PBMCs) and discovered that 17 of them had in vitro IgG memory for spike S1 antigen 60–90 days post infection, as previously indicated by in-cell ELISA [19]. The novelty of this research lies in the use of “in vitro” peripheral blood mononuclear cell (PBMC) cultures and metabolomics applied to these samples.

To obtain information on the construction of a memory B cell response for SARS-CoV2, Zarletti et al. [19] first adapted a simplified ELISPOT platform [38] to detect IgG produced in vitro for the spike S1 protein of SARS-CoV2, which has already been shown to be antigenic in humans and used as antigen in serological ELISAs [39]. The ability to secrete antibodies specific to antigens long after initial contact can be used to identify memory B cells and/or circulating plasma cells. Therefore, they can provide important information about previous exposure [40]. By utilizing serological ELISA analyses, the immune response can be screened for the presence of specific antibodies in the blood [41]. Coronavirus S1 protein is responsible for receptor binding and fusion with host cells and it is also linked to tropism and transmissibility [42]. The S1 protein was found to be a serological marker for COVID-19 [43]. Although it may require more work to increase the number of donors and sampling over time, cell ELISA may be more effective than conventional plasma or serum-based ELISA, particularly for IgGs that are undetectable in serological analysis [19]. This paper helps to assess immunization status by detecting S1 proteins, providing a clearer picture of the course of infection, and can be readily used in clinical trials. Moreover, immune responses are closely linked to metabolic programs [44,45]. Mass spectrometry (MS)-based omics is a reliable, unambiguous, and accurate approach that can be used to identify changes and the involvement of amino acid pathways, which are prerequisites for establishing a protective mechanism against SARS-CoV2 over time.

In this review, we scrutinized databases (Web of Science, Scopus, and Pubmed, 2020–2023) using general keywords associated with Metabolic Reprogramming in SARS-CoV2 and the immune system response. As a result of the large amount of information, we decided to select a collection of contributions that link to amino acid signatures in COVID-19 patients.

## 2. Amino Acid Metabolism

The most significant groups of metabolites include amino acids, which act as precursors for various major cellular components, such as proteins and nucleobases. Among the amino acids that make up proteins, nine cannot be synthesized from other compounds and must be obtained from food; these are also essential amino acids [46]. The human body can use amino acids ingested from food to synthesize proteins and other biomolecules, but they can also be oxidized to urea and carbon dioxide to produce energy through oxidative pathways. Their involvement in synthesizing proteins and metabolic regulators makes them an excellent marker for diseases [47]. Their chemical properties and compositions not only affect the structure and function of proteins, but also control the metabolic pathways associated with illness [48]. Furthermore, amino acids ensure the immune response against diseases by being used in the activation of T and B lymphocytes, natural killer cells, and macrophages [49]; in the cellular redox status, gene expression, and lymphocyte proliferation; and in the production of antibodies, cytokines, lymphokines, and cytotoxic substances [50].

Ali Ozturk et al. claimed that amino acids have an important role in metabolism and may affect COVID-19-related outcomes. In fact, the levels of many amino acids and their derivatives can change after SARS-CoV2 infection, and these changes are associated with disease severity. The authors evaluated the amino acid levels in the serum of patients with COVID-19 and found that alterations of several amino acids may be of prognostic value in the course of COVID-19 [48]. Recently, Shen et al. presented metabolomics and proteomics analyses of the serum of mild and severe COVID-19 patients indicating a massive suppression of amino acid metabolism in the sera of COVID-19 patients [51]. These catabolites were also observed in the early stages of SARS-CoV2 infection and they were correlated with future severe disease onset [43].

More studies have shown that changes in branched-chain AA (BCAA) metabolism are common in this pathological condition [50]; in the study of Junfang Wu et al. on COVID-19 patients, the inflammatory state led to an increase in BCAAs, aromatic AAs (AAs), and one-carbon related metabolites, which suggests a discrepancy in AA profiling in COVID-19 patients. In fact, most of the disturbed AAs recovered in convalescing subjects after 1 month post SARS-CoV2 infection [52]. Amino acid metabolic pathways central to leukocytes with in vitro IgG memory from SARS-CoV2-infected patients focused primarily on three major cycles (methionine cycle, arginine cycle, tryptophan cycle) involved in physiological recovery from viral infection [20,22]. The metabolomic fingerprint of unstimulated PBMCs analyzed up to 90 days after SARS-CoV2 infection is characterized by the dysregulation of some amino acids, especially methionine, tryptophan, and arginine metabolism (Figure 1).

### 2.1. Methionine Cycle

Methionine may modulate the assembly of SARS-CoV2 by interfering with the mechanism of RNA polymerase: SARS-CoV2 RNA-dependent RNA polymerase (RdRp) is used by SARS-CoV2 to replicate and transcribe genes, and methionine has the potential to disassemble SARS-CoV2 RdRp, which could be used to develop vaccines and therapies against COVID-19 [53]. The metabolism of a cell infected by the SARS-CoV2 virus is reshaped to fulfill the need for massive viral RNA synthesis, which requires de novo purine biosynthesis involving folate and one-carbon metabolism, suggesting that SARS-CoV2 takes over folate and one-carbon pathways for its intracellular replication [54].

SARS-CoV2 triggers antibodies against the spike S1 antigen, which is measurable in PBMCs starting 2 months after infection via in vitro assays [19]. In vitro cultured and unstimulated PBMCs had a significant alteration of metabolites related to the methylation cycle, as evidenced by the results of metabolomic analysis. S-adenosylmethionine (SAM) is a molecule that is produced in the body, consisting of the essential amino acids methionine and adenosine triphosphate. The molecule S-adenosylmethionine (SAM) is the methyl group donor [55]; under normal conditions, more than 90% of the total amount of SAM in mammalian cells is utilized for methylation reactions by AdoMet-dependent methyltransferases, during which SAM gives up its methyl group to various acceptors, including nucleic acids [56] (DNA, RNA), which play a crucial role in cell metabolism. The s-adenosyl-L-homocysteine (SAH), which is produced as a by-product of SAM-dependent methyl transfer reactions, is highly effective in inhibiting AdoMet-dependent methyltransferases and is broken down by S-adenosyl-L-homocysteine hydrolase into homocysteine and adenosine. In mammalian cells, by using 5-methyltetrahydrofolate cofactor (5-MTHF), homocysteine produced by this reaction can be remethylated to methionine and, thus, retained in the methylation cycle. Methionine is finally transformed into an SAM molecule by methionine adenosyltransferase (MAT). As a positive single-stranded RNA virus, SARS-CoV2 uses its genomic RNA both for translation and replication [57]. For proper RNA replication and translation, the cap of the viral RNA must be methylated [58]. It appears that two methylation sites are present in the viral RNA of coronaviruses; one site is required for replication and translation, and the other site may serve to allow the viral RNA to escape the host intracellular immune system, which would degrade the RNA without cap methylation [59]. The RNA cap of SARS-CoV2 is made up of a 7-methylguanosine attached to the 5′ nucleotide of the viral RNA through a triphosphate bridge. The cap is methylated at the N7 site of the guanosine, using SAM as a methyl donor, forming m7GpppN-RNA, mediated by NSP14 [60]. Then, by utilizing NSP16, the SAM-dependent 2′-O-methyltransferase attaches a methyl group to the ribose 2′-O site of the nucleotide to generate the cap (m7GpppNm-RNA) [61]. The RNA cap is involved in multiple aspects of gene expression, including boosting RNA stability, splicing, nucleocytoplasmic transport, and initiating translation, which is essential for viral RNA replication [62]. Therefore, S-adenosylmethionine (SAM) and S-adenosylhomocysteine (SAH) are indicators of global transmethylation and may play an important role as markers of COVID-19 severity. The risk of lung injury in patients with COVID-19 can be determined by the increased level of SAM, which is a marker of viral RNA capping’s necessity for its life cycle [63].

The ratio of SAM to SAH determines whether methyltransferase reactions can occur. The higher the ratio, the greater the methylation potential [64]; in contrast, if the ratio of SAM to SAH is low, methyltransferase reactions do not occur and coronavirus RNA is not methylated [65]. As a result, the virus cannot replicate and the viral genomes present in the cell are susceptible to degradation [57].

In this context, inhibition of the enzyme AdoHcy hydrolase may be used as a therapy against viral infections, as it indirectly limits the bioavailability of SAM and the methylation of the 5′-cap of the viral messenger RNA, as has already been found in the Ebola virus and the African swine fever virus [66,67]. At the same time, the strong increase in AdoHcy in IgGm+ PBMC suggests an inhibition of s-adenosyl-L-homocysteine hydrolase and, consequently, an imbalance in the SAM/AdoHcy ratio. Blocking viral mRNA caps could be a preliminary step in the development of antiviral therapies [19].

### 2.2. Arginine Metabolism

Arginine (Arg) is involved in many different biological processes and recent reports indicate that it could also play a crucial role in COVID-19 [68]. The amount of arginine available in the body has a significant impact on the normal immune system. Arginase-1 (Arg1), which has a pivotal role in immune cells, can be expressed in most of the myeloid cells, e.g., neutrophils and macrophages, and it is well known that it is an essential component of certain granulocyte subsets and can be released either locally or systematically during an immune response. The suppression of antiviral immune responses is associated with Arg1 [69]. Additionally, given the beneficial effects of arginine to significantly improve endothelial function, the control of long-term COVID-19 could be improved with arginine supplementation, as chronic inflammation and endothelial dysfunction are fundamental in COVID-19 progression [70,71]. Arg is a non-essential amino acid that is used by healthy humans to synthesize proteins and the urea cycle, and it is a precursor for various molecules, such as citrulline and nitric oxide (NO), which is a bioactive molecule with immunological and antimicrobial cytotoxic activity [72]. Arg can be provided in the diet or formed in certain cells through the complete or partial urea cycle. The synthesis of arginine as part of the urea cycle begins in the mitochondria, where carbamoyl phosphate condenses with ornithine through the action of ornithine transcarbamoylase to form citrulline, which leaves the mitochondria. In the cytosol, argininosuccinate synthase adds aspartate to citrulline, producing argininosuccinate, AMP, and pyrophosphate. The cleavage of argininosuccinate by argininosuccinate lyase yields arginine and fumarate. Arginine is then hydrolyzed by arginase to the final product, urea, and, simultaneously, ornithine is regenerated to re-enter the mitochondrion in exchange for citrulline via the ORNT1 transporter [73]. In addition, arginine can be transported from the extracellular space via the cationic amino acid transporter (CAT) and regenerated from citrulline, a product of nitric oxide synthase (eNOS), resulting in the citrulline–NO cycle in which nitric oxide is generated (NO) [74]. When arginase 1 (Arg1) degrades arginine in the urea cycle, it produces both urea and ornithine. When eNOS degrades it, there is a significant amount of NO and citrulline in the products [75]. NO is a substance produced by macrophages that are activated by either cytokines, microbial compounds, or both and is used to inhibit tumor growth both in vitro and in vivo [76]. In patients with IgG memory, arginine depletion via the urea cycle has become a substrate for the production of NO by iNOS, which plays a role in the first innate inflammatory immune response to viral infections. The production of NO is a characteristic of true cells of the immune system (dendritic cells, NK cells, mast cells, and phagocytic cells including monocytes, macrophages, microglia, Kupffer cells, eosinophils, and neutrophils) [77] and manages a variety of processes. The differentiation and proliferation of immune cells, proliferation and cell death, the production of cytokines and other soluble mediators, the expression of costimulatory and adhesion molecules, and the synthesis and deposition of extracellular matrix components are among these processes [78,79]. The progression of COVID-19 infection reduces the formation of NO, as infections lead to an increase in inflammatory cytokines in the peripheral circulation and trigger a strong cytokine storm [80]. In addition, inappropriately intense inflammation contributes to an imbalance of reactive oxygen species (ROS), leading to oxidative stress [81]. In the serum of patients with severe COVID-19, inflammatory cytokines and chemokines are found to promote excessive ROS production in mitochondria, ultimately leading to oxidative damage and cell death [82]. ROS also alter vascular tone by increasing intracellular calcium concentration and decreasing NO bioavailability [83]. Thus, NO has non-specific antiviral effects in various viral diseases and has been implicated in SARS-CoV2 virus replication [84]. In particular, Akaberi et al. confirmed that NO, which is derived from the NO-donor S-nitroso-N-acetylpenicillamine (SNAP), can delay or completely prevent the development of the viral cytopathic effect of SARS-CoV2 in treated cells and that the observed protective effect correlates with the degree of inhibition of viral replication [85]. In addition, inhaled NO can be used for COVID-19 prophylaxis and treatment in many phases, including the prevention of viral entry, symptom relief in critically ill patients, and supportive care in mechanically ventilated patients [86]. In the IgGm+ sample, activation of the urea cycle release of NO confirms that successful treatment and prevention options can be developed by manipulating this pathway. In addition to NO, arginine also plays a critical role in COVID-19 [68]. In COVID-19 patients, there is a high ratio of L-arginine to ornithine, which suggests a higher level of arginase activity [87]. In another study, it was found that the severity of COVID-19 was inversely correlated with plasmatic L-arginine levels [88]. In fact, a decrease in L-arginine bioavailability has been shown to lead to decreased T-cell response and function, and thus increased susceptibility to infection [89]. It is probable that the restoration of arginase (Arg1) is behind the accumulation of this amino acid in IgGm+. Arg1 has been found to be present in the cytoplasm and has a high expression level in the liver. In addition to its metabolic role in the hepatic urea cycle, it could also affect immune responses. Indeed, in humans, arginase is detected in peripheral blood mononuclear cells (PBMCs), and several studies show that Arg1 inhibits immunity to intracellular pathogens and suppresses T-cell-mediated inflammatory damage. In COVID-19 patients, an increase in Arg1 expression may be linked to an increase in viral load. Since Arg1 can limit the bioavailability of l-arginine, the inhibition of Arg1 can drive the recycling of l-citrulline to generate l-arginine for the production of NO, which contributes to the development of antiviral immunity in IgGm+.

### 2.3. Tryptophan Metabolism

According to Hikari Takeshita and Koichi Yamamoto, clinical studies have suggested that the kynurenine pathway of tryptophan metabolism is selectively enhanced in patients with severe COVID-19 [90]. Additionally, a study conducted by Gardinassi et al. revealed that inflammatory networks were heavily involved and genes implicated in tryptophan metabolism were upregulated in COVID-19 patients [91]. Tryptophan and its metabolites, including melatonin, can enhance the immune system and decrease inflammation in a variety of conditions [92]. The only way for humans to consume tryptophan (Trp) is through the diet, as it is an essential amino acid. Even though a small percentage of free Trp is employed for protein synthesis and the production of neurotransmitters like serotonin, over 95% of free Trp is used as a substrate for the kynurenine pathway of Trp degradation, which results in the generation of numerous bioactive metabolites in the immune response. The rate-limiting stage of the Kyn pathway involves the enzymatic transformation of Trp into N-formylkynurenine (NFK) by indoleamine 2,3-dioxygenase 1 (IDO1), IDO2, and tryptophan 2,3-dioxygenase (TDO) [93]. NFK is rapidly metabolized by kynurenine formamidase to L-kinurenine (L-Kyn). L-Kyn is an important metabolite that has potent immunoregulatory functions through its binding to the aryl hydrocarbon receptor (AhR) [94]. AhR binds to its response element XRE (or DRE) in the promoter of IL -6, thereby maintaining endogenous production of IL -6 and enhancing the inflammatory state [95]. In an inflammatory context sparked by cytokines such as interferon-γ, tumor necrosis factor α, and pathogenic infections such as influenza A virus or SARS-CoV2 infection [51], the activation of IDO-1 leads to the production of Kyn. The activation of Ahr by Kyn [96] regulates the immune response by suppressing the activity of natural killer cells, dendritic cells, monocytes, and macrophages, blocking the proliferation of T cells and promoting the proliferation of regulatory T cells [97]. In a recent study of COVID-19, some authors [91] reported increased activity of the tryptophan metabolic pathway, as evidenced by decreased TRP, increased KYN levels, and an increased KYN/TRP ratio [98,99], reflecting the activity of IDO [100]: acute inflammation rapidly triggers an “inflammatory storm” maintained mainly by the secretion of inflammatory cytokines, of which IL -6 is the most potent [101], enhancing the initial proinflammatory cytokine phase and suppressing the endogenous antiviral response. Several studies increasingly demonstrate that tryptophan and its metabolites, including melatonin, can reduce inflammatory responses and enhance the immune system [102,103]. Therefore, Kyn and other metabolites of the Kyn pathway have been proposed as potential biomarkers for COVID-19 [25]. In addition, IDO inhibitors may enhance the antiviral activity of COVID-19 [99]. In the post-infection stages (>60 days), when immune memory is responsible for protection against SARS-CoV2 reinfection, the BMC showed increased tryptophan levels, almost unchanged serotonin levels, and greatly decreased indole pyruvate levels, supporting the hypothesis of the restoration of kynurenine metabolism by attenuating the activity of IDO.

## 3. Conclusions

In conclusion, the contribution of this review was to gain a more comprehensive understanding of the biological mechanisms that underlie the immune response to post-COVID-19 infection and to gather current research that uses a metabolomics approach to investigate the association between amino acid metabolism and COVID-19.

Mass spectrometry measurements indicated that some altered metabolisms, such as the methionine cycle, arginine cycle, and tryptophan cycle, were impacted by COVID-19-induced changes in leukocytes. This review supports the idea that the metabolic profile changes observed in unstimulated PBMCs until after 90 days of SARS-CoV2 infection [20,22] could serve as a tool for evaluating disease, the modulation metabolism involved in innate and adaptive immunity, thus providing a new method for monitoring the risk of SARS-CoV2 reinfection. The analysis of these metabolites might represent a biomarker of the effective and long-standing antiviral activation of PBMCs.

## Figures and Tables

**Figure 1 diseases-12-00043-f001:**
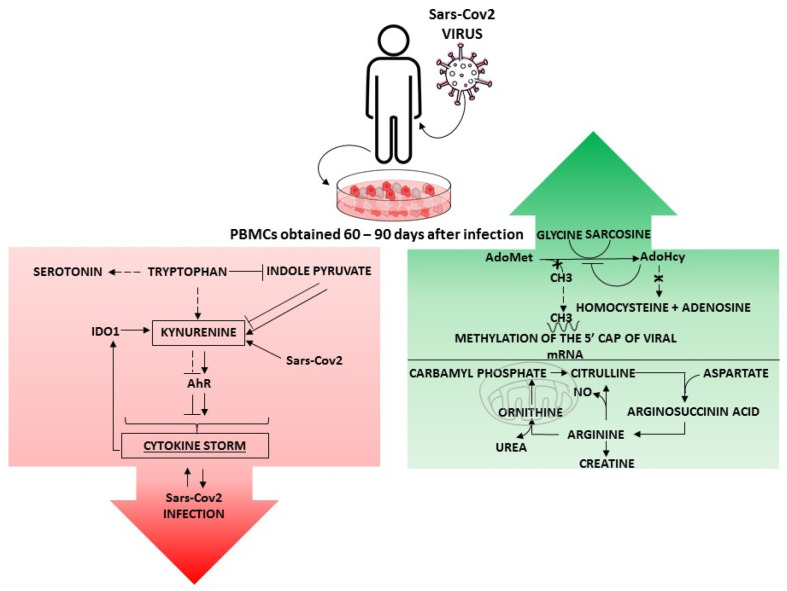
Differentially regulated signaling pathways in PBMCs exhibiting IgG antibody memory against SARS-CoV2. The green box shows the methionine and arginine metabolism pathways that were upregulated after infection, whereas the red box shows the tryptophan pathway that was downregulated in IgGm+ samples. Taken together, these pathways were involved in physiological recovery from viral infection and immune activity [20,22]. S-adenosylmethionine (AdoMet); S-adenosyl-L-homocysteine (AdoHcy); indoleamine 2,3-dioxygenase 1 (IDO1); aryl hydrocarbon receptor (AhR).

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
