# Peer review of "Amino Acid Metabolism in Leukocytes Showing In Vitro IgG Memory from SARS-CoV2-Infected Patients"

_diseases, 2024, doi:10.3390/diseases12030043_

Round 1

Reviewer 1 Report

Comments and Suggestions for Authors

This is reviewer has carefully gone through the review manuscript. This reviewer believes that there is significant scope of improvement of this review. Especially, authors need to provide adequate citations at many places, which are currently missing. The current draft reads fragmented and requires significant reshuffle and modifications in text. 

Specific comments: 

L33-34: "World Health Organization (WHO)". Abbreviate at first use.

L39: Reference 4 does not appear relevant. Please use relevant in-text references.

L40-41: Please provide more citations to support this. 

L42-43: There may be significant differences in symptoms caused by different variants. Please briefly discuss these differences among variants with relevant citations. Provided citation here is from the year 2020 and does not represent variations in symptoms caused by various variants. 

Same is true for the next sentence (reference 7). Please provide adequate citations while making claims. 

L64-65: Citations needed. Also please be more specific in writing. 

L68-72: Citations missing. 

L73-86: The entire paragraph severely lacking citations. Also, there is significant scope of improvement of the text here. 

L90-91: " alternations of amino acid metabolism over the course of SARS-CoV-2 infection has become one of the focuses in COVID 19 research." Authors made an interesting and strong statement without citations. The immediate following statement cited a paper published in 2007 (reference 19), well before COVID19 pandemic. The authors have generalised this section, however, the title of this section is : Metabolic rewiring in SARS-CoV2- infected patients". I would recommend authors write and cite specific studies that investigated SARS-CoV2 in recent past. 

L131-137: "To better understand the molecular mechanisms involved in the post-infection phases (> 60 days) during which immune memory is  responsible for protection against reinfection with SARS-CoV-2 COVID 19, we review key findings from the recent literature. In addition, our review also summarizes the current understanding of the relationship between amino acid metabolism and the immune system response." This appears to be introductory paragraph and must be moved to an appropriate place. 

L139-156: This entire paragraph reads like a book chapter and is very basic. It does not connect with the subject discussed here. Also, it severely lacks citations. 

L165: Reference 30 is not related to COVID 19. 

L170-177: Citations missing. 

On what basis Figure 1 was drawn? Citations missing.

Please replace references 34, 35, 36, 37 with recent and relevant references citing SARS-CoV or SARS-CoV2. 

L258-265: Citations missing.

The sections describing methionine and arginine metabolism discuss general aspects of metabolism with limited information to the subject of this review- SARS-CoV2. There is a scope of significant modification and I recommend authors focus on the information related to SARS-CoV2 instead of digressing from the main subject.

I recommend authors briefly mention how they collected relevant literature that was used to write this review. This is particularly important because if authors ignored more relevant and recent studies then their conclusions may be biased. 

The current version is also lacking a conclusions section. Please sum up the conclusions from this study in one paragraph. 

Comments on the Quality of English Language

Minor editing of English may be needed. 

Author Response

First of all, we would like to thank you the Reviewer for the encouraging comments on the manuscript. We hereby submit a revised version, modified according to the suggestions of the reviewer. All changes are marked with red text. We sincerely hope that this deeply revised version of the manuscript will succeed in meeting the Reviewers’ expectations.

Comment:L33-34: "World Health Organization (WHO)". Abbreviate at first use.

Author’s reply: As Referee suggestion, World Health Organization (WHO) was abbreviated (Line 37)

Comment: L39: Reference 4 does not appear relevant. Please use relevant in-text references.

Author’s reply: As required, the reference has been replaced with a more relevant paper “Huang C, Wang Y, Li X, et al. Clinical features of patients infected with 2019 novel coronavirus in Wuhan, China. Lancet. (2020) ;395(10223):497-506” (Line 43).

Comment: L40-41: Please provide more citations to support this.

Author’s reply: We accordingly added more citations to support that loss of smell and taste represents an early clinical marker for COVID-19. (Line 45) (References 5, 6, 7, 8)

Comment: L42-43: There may be significant differences in symptoms caused by different variants. Please briefly discuss these differences among variants with relevant citations. Provided citation here is from the year 2020 and does not represent variations in symptoms caused by various variants. Same is true for the next sentence (reference 7). Please provide adequate citations while making claims.

Author’s reply: We have carefully taken under consideration the reviewer's comment and we added a few comments that take into account the variations in symptoms caused by various variants. All changes are marked with red text. (Lines 48-58)

Comment: L64-65: Citations needed. Also please be more specific in writing

Author’s reply: Many thanks for the observation. Indeed, it appears a bit confusing. In the new version of the manuscript the sentence has been clarified. (Lines 69-76)

Comment: L68-72: Citations missing.

Author’s reply: References were accordingly added.(Lines 90-91)( References 23, 24, 25)

Comment: L73-86: The entire paragraph severely lacking citations. Also, there is significant scope of improvement of the text here.

Author’s reply: The authors are grateful to the reviewer for his/her constructive suggestions. So we revised our introduction adding more information (adding also specific references) in line with the purpose of the review. All changes are marked with red text. (lines 96-98; 105-107) (References 26, 27, 28, 30).

Comment: L90-91: " alternations of amino acid metabolism over the course of SARS-CoV-2 infection has become one of the focuses in COVID 19 research." Authors made an interesting and strong statement without citations. The immediate following statement cited a paper published in 2007 (reference 19), well before COVID19 pandemic. The authors have generalised this section, however, the title of this section is : Metabolic rewiring in SARS-CoV2- infected patients". I would recommend authors write and cite specific studies that investigated SARS-CoV2 in recent past.

Author’s reply: To comply with this Reviewer’s comment, the revised manuscript now provides a more detailed explanation of the connection between amino acid metabolism and COVID-19. (Lines 118-134) (References 31, 32, 33, 34, 36, 37, 38)

Comment: L131-137: "To better understand the molecular mechanisms involved in the post-infection phases (> 60 days) during which immune memory is responsible for protection against reinfection with SARS-CoV-2 COVID 19, we review key findings from the recent literature. In addition, our review also summarizes the current understanding of the relationship between amino acid metabolism and the immune system response." This appears to be introductory paragraph and must be moved to an appropriate place.

Author’s reply: The authors have acknowledged the value of this reviewer's suggestions and want to thank them. Therefore, in the revised section, we have added some information about the methodological strategy used to build the review. (Lines 172-177)

Comment: L139-156: This entire paragraph reads like a book chapter and is very basic. It does not connect with the subject discussed here. Also, it severely lacks citations.

Author’s reply: We accepted the referee's suggestion and improved the paragraph by incorporating more information about the influence of amino acid metabolism on COVID-19 disease. We believe that this suggestion enhances the quality of the manuscript. (Lines 195-200; 213-218) (References 44, 47-49, 51, 52)

Comment: L165: Reference 30 is not related to COVID 19.

Author’s reply: We apologize for the misleading reference. According to the referee's request, we have changed it. (Line 220) (Reference 51)

Comment: L170-177: Citations missing.

Author’s reply: The information that was requested has been included. (Lines 229) (References 20, 22)

Comment: On what basis Figure 1 was drawn? Citations missing.

Author’s reply: The figure provides a graphic view of the data from our previous publications regarding the metabolic profile of peripheral blood mononuclear cells (PBMC) cultured in vitro 60-90 days after COVID-19 infection. The figure reveals the significant metabolites that have changed in leukocytes with SARS-CoV-2 IgG antibody memory and their metabolic profile. We included references to manuscripts that were mentioned (Line 240) (References 20, 22)

Comment: Please replace references 34, 35, 36, 37 with recent and relevant references citing SARS-CoV or SARS-CoV2.

Author’s reply: Authors want to thank for the suggestion but our point of view is not aligned with that of this reviewer.

Comment: L258-265: Citations missing.

Author’s reply: References were accordingly added. (Line 317) (Reference 69)

Comment: The sections describing methionine and arginine metabolism discuss general aspects of metabolism with limited information to the subject of this review- SARS-CoV2. There is a scope of significant modification and I recommend authors focus on the information related to SARS-CoV2 instead of digressing from the main subject.

Author’s reply: The authors are grateful for the referee's suggestions. However the purpose of this paragraph is to describe the metabolic processes of methionine and arginine in order to strengthen the connection between these processes and COVID-19 infection.  Furthermore, the documents we frequently mention specifically address the closely connected relationship between amino acids and post-COVID-19 infection.

Comment: I recommend authors briefly mention how they collected relevant literature that was used to write this review. This is particularly important because if authors ignored more relevant and recent studies then their conclusions may be biased.

Author’s reply: We have carefully taken into consideration the reviewer's comment and added phrases describing the procedures applied to decide on the inclusion of articles. (Lines 172-177)

Comment: The current version is also lacking a conclusions section. Please sum up the conclusions from this study in one paragraph.

Author’s reply: In order to comply with this Reviewer’s constructive comment, we have added the conclusion paragraph. (Lines 462-476)

Reviewer 2 Report

Comments and Suggestions for Authors

Estimated Editors in "DISEASES",

I've been invited to review the paper from Fanelli et al. on the impact of SARS-CoV-2 infection in Leukocytes' metabolism of AA. 

According to the Authors, who collected and systematized available evidence about this specific topic, SARS-CoV-2 infection elicits a series of sustained changes in the AA metabolism in leukocytes. Albeit only partially explained, in particular when dealing with the maintainance of this derangement, these evidence may contribute to our understanding of some effects of SARS-CoV-2 infection on human health, not only in the acute but also in chronic stages.

From my point of view, the present version of the paper (that is a revision of a previous version, not assessed by the present reviewer) could be accepted for publication after some minor adjustments.

First of all: the paper is well written and organized. Still, Authors should provide a conclusive Discussion and Conclusion section where the evidence about Arg, Trp, and Met are summarize and recollected in order to provide a common end message, that could include some tentative summary explanations for the sustained derangement of the AA metabolism after the clearance of SARS-CoV-2 infection.

Second, very minor: please be careful with acronyms SARS-CoV-2 (not SARS-Cov2) and COVID-19 as both are not consistently spelled across the paper.

Author Response

First of all, We are very glad that the Reviewer positively evaluated our manuscript. We hereby submit a revised version, modified according to the suggestions of the reviewer. All changes are marked with red text. We sincerely hope that this revised version of the manuscript will succeed in meeting the Reviewers’ expectations.

Comment: First of all: the paper is well written and organized. Still, Authors should provide a conclusive Discussion and Conclusion section where the evidence about Arg, Trp, and Met are summarize and recollected in order to provide a common end message, that could include some tentative summary explanations for the sustained derangement of the AA metabolism after the clearance of SARS-CoV-2 infection.

Author’s reply: The authors would like to thank this Reviewer for his/her kind comments. We are very glad that the Reviewer positively evaluated our review. Moreover, we have carefully taken under consideration the reviewer's comment and we have added the conclusion paragraph. (Lines 462-476)

Comment: Second, very minor: please be careful with acronyms SARS-CoV-2 (not SARS-Cov2) and COVID-19 as both are not consistently spelled across the paper.

Author’s reply: We would like to apologize for these slight imprecisions that we have now rectified.

Reviewer 3 Report

Comments and Suggestions for Authors

Thank you for sharing your article on amino acid metabolism in leucocytes expressing IgG memory from SARS-CoV2 patients.

The literature review reads really well. What I am missing is a methodological section stating which databases were accessed and when you accessed them last time, including the search terms/strategy applied. Which procedure(s) did you apply to decide on the inclusion/exclusion of articles detected through your search? Also, I miss a conclusion section, which should round off every article. 

L139: You stated that the most significant groups of metabolites include amino acids. Consequently, what is your rationale of presenting only amino acids? 

Comments on the Quality of English Language

Please see above. 

Author Response

First of all, the authors would like to thank this Reviewer for his/her kind comments. We are very glad that the Reviewer positively evaluated our manuscript and for allowing us to improve the quality of the review. We hereby submit a revised version and We believe that after the inclusion of the Reviewer’s proposals, our manuscript is truly improved

Comment:  The literature review reads really well. What I am missing is a methodological section stating which databases were accessed and when you accessed them last time, including the search terms/strategy applied. Which procedure(s) did you apply to decide on the inclusion/exclusion of articles detected through your search? Also, I miss a conclusion section, which should round off every article.

Author’s reply: We would like to thank the Reviewer 3 for the encouraging comments on the manuscript. We submit a revised version, modified according to the suggestions of the reviewer, adding phrases describing the procedures applied to decide on the inclusion of articles and the conclusion paragraph. (Lines 172-177; 458-473)

Comment:  L139: You stated that the most significant groups of metabolites include amino acids. Consequently, what is your rationale of presenting only amino acids?

Author’s reply: The authors thank the reviewer for the appropriate comment. The aim of this review was to investigate metabolomics concerning of  COVID-19, and starting by our recent papers [20, 22], which have shown that the most metabolic signature of COVID-19 was represented by amino acids, we planned to emphasize the importance of these metabolites in peripheral blood mononuclear cells (PBMC) after COVID-19 infection to understand the metabolism involved in innate and adaptive immunity, and propose innovative approaches to monitor the risk of SARS-CoV2 reinfection.

Round 2

Reviewer 1 Report

Comments and Suggestions for Authors

Dear Authors,

Thanks for the revision. Please correct occassional typos, such as Line 136: we provide.

Comments on the Quality of English Language

Minor editing may be required. Please read the entire text carefully and correct the typos.